# Supporting Circular Economy Principles by Recycling Window Frames into Particleboard

**DOI:** 10.3390/ma17164132

**Published:** 2024-08-21

**Authors:** Anita Wronka, Grzegorz Kowaluk

**Affiliations:** Institute of Wood Science and Furniture, Warsaw University of Life Sciences—SGGW, Nowoursynowska St. 159, 02-776 Warsaw, Poland

**Keywords:** particleboard, circular economy, recycling, window frames, furniture

## Abstract

The aim of the study was to identify limiting factors for reusing wood through the recycling of window frames by conducting research under fully controlled conditions. The research involved manufacturing new window frames, seasoning them, and then shredding them into wood particles to prepare a three-layer particleboard. The proportion of wood particles in recycling was 0, 5, 10, 25, 50, and 100 parts by weight of the manufactured particleboard. Mechanical property tests were conducted: modulus of elasticity (MOE) and modulus of rupture (MOR), internal bond (IB), screw withdrawal resistance (SWR), and physical properties: density profile (DP), thickness swelling (TS) after water immersion, water absorption (WA), as well as formaldehyde emission and total volatile organic compound (TVOCs) tests. The research indicates a significant potential for utilizing wood from this sector of the wood industry, particularly considering variants with a higher proportion of recycled wood. MOR and MOE results are most promising for variants above 50 parts by weight of recycled wood. Based on the results obtained, it is clear that the production process should be improved or the raw material modified to enhance the internal bonding of particleboard, as these results were the weakest. Thus, recycled wood from window joinery has the potential to be reincarnated as particleboard, which continues to be widely used in their production.

## 1. Introduction

The global shift towards resource recovery is becoming increasingly prevalent, with the wood industry also experiencing this trend. This movement is not merely coincidental; it is driven by the pressing challenges of climate change, population growth, and the resulting increase in consumption, which collectively contribute to environmental degradation. The circular economy offers a robust solution to the demand for sustainable production by optimizing the use of raw materials and minimizing carbon footprint, while reducing the emission of harmful substances [1,2,3,4]. Recycling wood waste in construction helps reduce deforestation, lower carbon emissions, and mitigate biodiversity loss, making it an essential strategy for environmental conservation [5]. This approach fosters multi-level growth within the industry by encouraging industrial collaboration, where producers exchange by-products [6] to ensure they are disposed of in the most efficient way possible [7,8]. The circular economy model emphasizes the importance of managing energy recovery from waste, especially from the wood sector, to reduce raw material consumption and promote economic growth [9]. A common practice has been to burn wood waste for energy, but this is not the most efficient use of a raw material that often still holds significant value [10,11], when it comes to recycling and disposal of used wood and wood residues. An alternative to burning wood is generating energy from it in a steam generator, producing pellets [12], or producing biofuel, such as ethanol [13]. Wood composites can be recycled through various methods, including mechanical, thermo-hydrolytic, hydrothermal, and chemical processes, or a combination of these techniques [14,15]. The biggest challenge in wood recycling is the presence of chemical contaminants, such as wood preservatives, paints, and adhesives, which are difficult to remove mechanically from wood waste. For this reason, wood waste is often stored rather than recycled. Disposing of wood waste is challenging due to organic and inorganic preservatives, which significantly limit recycling options [16,17].

Every year, European countries generate an estimated 19 million tons of rubbish, 60% of which ends up in landfills, from various commodities such as furniture, mattresses, upholstery, textiles, and plastic garden products. A study shows that 2030 wood supply may not satisfy European demand [18]. Analyzing the amount of post-consumer wood waste, a 2013 study in Poland estimated the potential volume of post-consumer wood waste to be 6.5 million cubic meters, with the construction sector being the primary contributor, responsible for 64% of this volume. The composition of wood waste, presented in percentage terms, is as follows: sawn wood (approximately 83%), wood-based materials (9%), and roundwood (8%). Additionally, used wooden windows and doors comprise 15% of the waste [19]. The cost of fresh wood material is increasing, and a future shortage is anticipated, highlighting the potential economic impact of Poland’s annual generation of 3 million cubic meters of post-consumer wood waste [20]. Wood byproducts were also examined. In 2015, the potential volume of wood by-products generated in Poland’s wood sector surpassed 13 million cubic meters, with 61% used directly within the industry for production and energy [21]. 

One example explores the use of 30-year-old window frames for producing three-layer particleboard. These studies utilized window frames made of (*Pinus sylvestris* L.), including both painted and cleaned frames. The research investigated how frame density, adhesive quantity, and paint presence affect the particleboard’s strength properties. Contaminants such as nails and screws were removed, and the frames were shredded and sorted appropriately. The findings revealed that removing paint from the wood particles notably improved the mechanical properties of the particleboard. Although paint enhanced surface roughness, it negatively impacted mechanical strength. Panels from painted particles exhibited higher density but reduced strength due to less effective adhesive bonding. The study demonstrated that waste from Scotch pine window joints holds substantial potential as a raw material for particleboard production. It highlighted that while increasing density and adhesive amount enhances particleboard properties, the presence of paint and the absence of hydrophobic substances can adversely affect properties such as thickness swelling [22]. The thermal performance of a window is greatly affected by the material used for its frame. With the growing emphasis on sustainable design, window frame materials must demonstrate superior environmental performance to be considered truly sustainable. Consequently, a comprehensive performance metric is needed to evaluate window frame materials effectively. This study analyzed three identical aluminum, polyvinyl chloride (PVC) frames and wood. Their thermal performance was initially assessed and compared using a heat transfer model. Subsequently, the carbon footprint of each material was calculated for a one-square-meter window area with comparable thermal performance. The results revealed that wood window frames excelled over aluminum and PVC in thermal and environmental performance. However, aluminum frames had a significant environmental impact and lower thermal performance. This study thoroughly analyses window frames, considering their environmental and thermal performance [23]. Window frames are classified as bulky waste, complicating their collection and recycling compared to routine waste materials. The collection process for window frames is less efficient than everyday waste. These frames serve as structural support, providing form and stability to each window. As lifestyle changes and economic conditions drive the demand for inexpensive, quickly replaceable products, bulky waste, including window frames, has become a relatively small but steadily increasing waste stream in many countries [24].

Recycled wood can be used as a raw material in producing wood-based composites, such as particleboard, fiberboard, and structural panels, resulting in high-value products [25]. Unfortunately, the literature offers limited examples of recycling window frames, but there are instances of recycling other wood–plastic composites that could inspire further research. One notable example is the hydrothermal processing of particleboard, where the recovered wood material is reused to manufacture new particleboard. However, this process tends to degrade the raw material, which negatively impacts its strength properties [15]. It is important to note that, depending on atmospheric conditions, and the color and quality of the coating, any varnish that is not removed during recycling can negatively impact the strength properties of the manufactured wood materials. However, it has also been demonstrated that a suitable coating can help maintain the highest wood quality, even after many years of use [26,27,28]. Reclaimed wood from demolition can be successfully repurposed for structural use, with promising applications in creating engineered wood products (EWPs) like cross-laminated timber (CLT) and glulam. This approach minimizes waste and contributes to the sustainability of wood construction by incorporating recycled materials into advanced building components [29]. Recent projects have faced challenges, with unsatisfactory outcomes primarily due to wood species’ diversity and difficulties in sorting materials. Wood sourced from furniture, window frames, and moldings presented significant obstacles to maximizing the use of recycled materials. As a result, subsequent efforts have concentrated on developing more effective methods for sorting raw materials for recycling [30,31,32]. A notable advantage of window frames is their relatively large dimensions, which makes them ideal for shredding and repurposing in applications such as particleboard production [33]. Egger is a company that exemplifies the sustainable use of wood waste by producing particleboard. To ensure a steady supply of post-consumer wood, the company has established collection points and collaborates with other similar facilities. This approach allows them to replace approximately 45% of their original raw material with recycled wood. Additionally, Egger actively promotes wood recycling through public awareness campaigns and has hosted open days at its factory to educate the public about the importance of recycling wood [34].

Recycling wooden window frames supports a closed-loop economy by enabling the high-quality reuse of critical components after the window’s useful life has ended. Guidelines for designing wood windows focus on allowing the high-quality recycling of major components once the window has reached the end of its useful life. These guidelines recommend materials, connections, structural design, and component geometry to facilitate efficient and cost-effective recycling processes [35]. Analyzing the life-cycle impact of recycling various building components and materials allows the industry to understand the environmental benefits of reuse and recycling, thereby supporting the adoption of sustainable construction practices [36]. Therefore, it is crucial to keep the state of the art up to date, given the rapid growth of the wood coatings industry. This fast-paced development necessitates continuous research and innovation.

In conclusion, while shredding window frames may appear to be a straightforward solution for recycling, scientific reports highlight numerous challenges, mainly due to the variety of grades and coatings present [37,38]. This research aims to identify the key factors that limit the recycling of window frames to produce three-layer particleboards by conducting studies under fully controlled conditions, where the raw materials, processes, and coatings are precisely known—factors that typically complicate the recycling of such materials.

## 2. Materials and Methods

### 2.1. Materials

Industrial particles, pine (*Pinus sylvestris* L.) and spruce (*Picea abies* (L.) H.Karst) with about 3% moisture content (MC), intended for use in face and core layer particleboard production (reference raw material) and obtained from the Research and Development Centre for Wood-Based Panels Sp. z o. o. in Czarna Woda, Poland, were used. Silekol S-123 (Silekol Sp. z o. o., Kędzierzyn—Koźle, Poland) of about 66% dry content (EN 827 2005 [39]) urea-formaldehyde (UF) industrial resin was used, with ammonium nitrate water solution as a hardener, to reach a curing time of the gluing mass in 100 °C of about 86 s). The imitation post-consumer window frames were prepared from pine (*Pinus sylvestris* L.) sawn wood in the laboratory and then painted—with clear varnish and white paint. Two types of wooden window surface professional finishing systems, transparent and white, including acrylic primer, base, and face coats, delivered by international brands, were used for surface finishing. We do not publish the provider’s name to avoid connecting the results with the coating producer.

### 2.2. Preparation of Panels

Following the production process for the window frames, a waterproofing layer, a primer, an interlayer, and an appropriate top-coat were applied. The application was carried out by the manufacturer’s recommendations. The window frames prepared this way were seasoned for one year and then re-milled using a laboratory knife milling machine. This laboratory prototype was provided by the Research and Development Centre for Wood-Based Panels Sp. z o. o. in Czarna Woda, Poland, and equipped with knives and counter-knives similar to those in industrial milling machines used for particleboard production. During the particles’ resination, the glue was sprayed over the particles with an air gun while they were being blended in a laboratory mixer. The following resination was applied: 12% and 10%, face and core layer, respectively. No hydrophobic agents were added. The panels had a nominal density of 670 kg m^−3^, were 16 mm thick, and 32% of the mass was from the face layers. The share of post-consumer particles was 5, 10, 25, 50, and 100 parts by weight. The manually formed mats were initially pre-pressed on a hydraulic press (ZUP-NYSA PH-1P125) at room temperature for 30 s under the maximum unit pressure of 0.9 MPa, then pressed in a hot (200 °C) hydraulic press (AKE, Mariannelund, Sweden) with a press factor of 20 s mm^−1^ of the nominal thickness of the panel under the maximum unit pressure of 2.5 MPa. The pressing regime is presented in Figure 1. Before the tests, the created boards were conditioned at 20 °C and 65% ambient air humidity to achieve a constant mass, and all samples were calibrated (sanded to the nominal thickness) before testing.

### 2.3. Characterization of the Elaborated Panels

The initial phase of the study involved the prepared window frames, with the determination of coating hardness carried out according to ISO 1522 (2022) [40] standards. Subsequently, the window frames were processed to produce wood particles. The resulting particles underwent testing for bulk density and fractional composition via sieve analysis. The following mechanical and physical characteristics were assessed in this study using European standards (where applicable): density [41], bending strength (modulus of rupture—MOR) and modulus of elasticity (MOE) [42], internal bonding (IB) determined according to EN 319 [43], screw withdrawal resistance (SWR) [44], and water absorption (WA) and thickness swelling (TS) after 2 and 24 h of immersion in water [45]. All mechanical properties were tested on a computer-controlled universal testing machine (Research and Development Centre for Wood-Based Panels Sp. z o.o. Czarna Woda, Poland). For every test of mechanical and physical parameters, a minimum of 10 samples of each type of panel were used. Test specimens with dimensions of 50 mm by 50 mm were used, and they were analyzed using a Grecon DA-X measuring device (Fagus-GreCon Greten GmbH & Co. KG, Alfeld/Hannover, Germany) using direct X-ray densitometry scanning panel thickness in a 0.02 mm sample step to obtain the density profile (DP). After analyzing three samples of each test variant, a representative density profile was chosen for each panel type to be further analyzed. Where possible, the selected results were referenced to the European standard [46].

The emission of formaldehyde (HCHO) and total volatile organic compounds (TVOC) was measured on three samples per tested panel using the method described in [47]. Three samples from the reference variant and the variants made entirely from recycled materials were used for this study. 

### 2.4. Statistical Analyses

Analysis of variance (ANOVA) and t-test calculations were used to test (α = 0.05) for significant differences between factors and levels using the IBM SPSS statistic base (IBM, SPSS 20, Armonk, NY, USA). A comparison of the means was performed, and the homogenous and non-homogenous groups were collected in Table 1. Where applicable, the mean values of the investigated features and the standard deviation indicated as error bars have been presented on the plots as error bars.

## 3. Results and Discussion 

### 3.1. Fraction Share and Bulk Density of Particles

Figure 2 shows the fractional composition of particles from the milling of white-painted and clear-painted window frames compared to particles conventionally used in the particleboard industry. For industrial particles, most fractions fall within the 1–4 mm range. In contrast, particles obtained from reclaimed wood window frames are reduced during shredding, with most fractions ranging from 0.25–4 mm, particularly for those with white paint. A similar trend is observed with particles from clear varnish-painted window frames; however, these contain a higher proportion of fine fractions than white-painted particles. The higher density and brittleness of varnishes compared to wood likely contribute to the shortening of the particles. A similar trend of particle shortening was observed in studies where particleboards were subjected to repeated milling. These particleboards contained laminate, which has a high density similar to varnish. Additionally, the adhesive content increased with each successive batch of particleboard made from particleboard residues. Once cured, this adhesive also has a higher density than wood [48]. Consequently, there was a significant shortening of wood fibers, as confirmed by the studies of Wronka and Kowaluk [49].

Figure 3 illustrates the bulk density of various raw materials. The industrial particles, free from contamination, recorded the lowest bulk density (145 kg m^−3^), a finding consistent with the literature [49]. The blue bar represents the bulk density values of window frame mill particles painted with clear paint (177 kg m^−3^). This value is higher than industrial particles but lower than the bulk density of particles from milled window frames painted with white paint (192 kg m^−3^). The differences in these values may be attributed to the varying compositions of the coatings used. White paint, which contains a filler, likely contributes to the increased bulk density.

### 3.2. Relative Hardness and Density Profile

Figure 4 shows the relative hardness of the coatings used. The surface with clear varnish exhibited a higher relative hardness of 0.15, whereas the surface coated with white varnish had a relative hardness of 0.11. This information is crucial when considering the fractional composition of the particles after milling. More complex coatings tend to be more brittle, which can increase the proportion of smaller fractions, specifically those less than 2 mm, or even dust. The values obtained are consistent with other studies [50]. Another study investigated the effect of temperature on various coatings (varnishes were applied to the heated wood), including cellulose lacquer, synthetic varnish, polyurethane varnish, and water-based varnish. The initial heat treatment of the wood did not contribute to the relative hardness of the varnishes used [51]. Varnished wooden surfaces are exposed to various external variables depending on their application. To avoid economic loss, it is critical to select varnishes that provide optimal performance matched to the specific area of application [52]. Studies report that the enrichment of the varnish with TiO_2_ and Al_2_O_3_ nanofillers promotes a reduction in the thickness of the surface layer and thus contributes to the hardness of the coating and the ability to absorb UV radiation [53].

The diagram in Figure 5 shows the relative hardness for the clear varnish and the white paint used in the tests. The blue line shows the density profile of the white paint, which has a lower density and does not penetrate the wood as deeply as the clear varnish (orange line), which may be due to the presence of fillers in the paint. The lack of fillers in the paint makes the paint penetrate the wood better, which can affect some test results, such as water absorption or bulk density. Paint layer thickness can vary greatly depending on solvent evaporation, surface tension gradients, and paint application processes [54], and also, the temperature of the liquid top coating can impact the viscosity of the coating, which therefore affects the dry film thickness on wood surfaces [55]. In addition, the durability of the tested coating was discovered to be better for spruce wood, demonstrating that different wood species can influence the total life and performance of the paint covering [56].

### 3.3. Modulus of Rupture and Modulus of Elasticity

The results for MOR are presented in Figure 6. The MOR value was observed to increase with the amount of recycled content. When analyzing the MOR results for the variant using window frames with white paint, the highest MOR value was recorded for the variant made entirely from recycled particles at 14.5 N mm^−2^, which is slightly lower than the reference variant at 15.3 N mm^−2^. The weakest performance was observed in the variant where recycled particles accounted for five parts by mass, with an MOR value of 10.1 N mm^−2^. For the clear varnish wood, the highest MOR value was achieved by the variant made entirely from recycled particles at 16.8 N mm^−2^, exceeding the reference value of 15.3 N mm^−2^. The weakest performance in this category was seen in the variant containing 10 parts by mass of reclaimed wood, with an MOR value of 10.6 N mm^−2^. The lacquer increases the surface hardness of the particles, which can lead to an enhancement in the mechanical strength of the particleboards. Higher hardness can improve the MOR, as the lacquer coating may serve as an additional protective layer, enhancing resistance to breaking. A similar study investigated the effect of density and bonding on the properties of particleboard made from milled window frames, recording comparable MOR values for boards with a density of 700 kg m^−3^ [22]. Unfortunately, not all of the produced variants met the MOR standard values. The most promising results were observed in particleboards made entirely from recycled material, and the highest values were recorded for particles from lacquered window frames. Depending on the origin of the raw material used, the density of the particle board may need to be individually tailored, as it largely determines its strength regarding various mechanical properties, including MOR [57]. 

The MOE is shown in Figure 7. The trend is similar to that observed for the MOR: as the recycled wood content increases, the MOE also increases. However, unlike the MOR results, all variants met the standard for MOE. For particleboard made from window frame particles painted white, the highest MOE value was recorded for the variant made entirely from the alternative raw material at 3032 N mm^−2^, which is lower than the reference value of 3212 N mm^−2^. The lowest MOE value was observed in the panel where alternative particles accounted for 10 percent, with an MOE of 2251 N mm^−2^. In the case of specimens made from window frames with clear lacquer, the highest MOE value was also recorded for the variant made entirely from alternative particles, reaching 3665 N mm^−2^, which exceeds the reference value. Conversely, the lowest MOE was obtained in the variant where the recycled raw material constituted 10 parts by mass, recording a value of 2241 N mm^−2^. Research confirms that recycled wood can reduce parameters such as the MOE. Nevertheless, the results met the EN 312 standards [58]. During the preparation of the particleboard, dust fractions can also form, for example, when mixing the raw material with the adhesive. Studies confirm that the presence of dust fractions in the core layer of the board can contribute to the weakening of MOE parameters [59].

### 3.4. Internal Bond

The results of the internal bond test are depicted in Figure 8. Notably, most variants failed to meet the minimum IB requirements. The IB values are as follows: for particleboards made from white-painted window frames, the highest value was recorded for the variant made entirely from alternative particles, at 0.47 N mm^−2^, which is lower than the reference value of 0.61 N mm^−2^. Only this variant met the standard requirement for IB, which is 0.35 N mm^−2^. For samples made from window frames coated with clear lacquer, the highest IB value was also recorded for the variant made entirely from recycled particles, achieving 0.89 N mm^−2^, which was higher than the reference value. Similarly to the boards with white paint, only this variant from the series of samples with clear lacquer met the standard requirements. The weakest IB performance in both cases was observed in samples where the alternative raw material constituted 10 parts by weight—0.09 N mm^−2^ (white paint) and 0.13 N mm^−2^ (clear varnish). Thermally modified wood (TMW) was tested for recycling into a three-layer particleboard. Boards were produced with recycled wood contents of 0, 20, 50, and 100. As the TMW content increased, the IB value decreased [60]. The thermo-modification of wood mentioned above can adversely affect the wettability and, thus, the sticking of wood particles, which is confirmed by similar studies using TWM [61]. 

For IB, the size of the particles is crucial, especially in the core layer. As observed during the milling of window frames, wood particles tend to shorten compared to conventional particles. This is partly due to the presence of finishes, particularly clear varnish, which penetrates the wood more deeply and contributes to its increased brittleness. This brittleness leads to a more pronounced reduction in particle size [Figure 2], not only during milling but also during the process of gluing the particles.

### 3.5. Screw Withdrawal Resistance 

Screw withdrawal resistance is shown in Figure 9. The lowest SWR values were recorded for variants where the alternative crude was 5 (106 N mm^−1^), 10(86 N mm^−1^), and 25 (111 N mm^−1^) parts by weight for particleboards made from white-painted window frames. The highest value was recorded for the variant made entirely from alternative particles—139 N mm^−1^. For the particleboard made of particles with clear varnish, the highest value was 165 N mm^−1^; the values obtained were even higher than for the reference board. The lowest SWR value for particleboards made from window frames with clear varnish was recorded for the variant where recycled particles accounted for 25 parts by weight—97 N mm^−1^. The density of the substance under test determines SWR, which can significantly increase as density increases [62,63]. In contrast, other studies report that recycled wood contributes to lower SWR values [47,64]. Still, it is worth bearing in mind that the recycled raw material and, above all, its form does matter, so it is essential to use more significant fractions, especially in the core layer [65].

### 3.6. Water Absorption and Thickness Swelling

The water absorption diagram is shown in Figure 10. Water absorption decreased as the post-treatment wood content increased after 2 and 24 h. The particleboards made entirely of post-consumer wood and coated with clear varnish exhibited the best dimensional stability. After 2 h, for the variant with 0% alternative raw material (varnish one), the water absorption was 87.4%; and, as the proportion of alternative raw material increased, the water absorption decreased, and for 100%, WA significantly decreased to 75.5%. After 24 h, for the variant with 0% alternative raw material, the water absorption is 94.1%. As the proportion of alternative raw material increases, the water absorption decreases to 87.1% with 100% alternative raw material. For samples made from white-painted window frames, water absorption (WA) after 2 h was 87.4% for the variant with 0% alternative raw material. As the proportion of alternative raw materials increased, water absorption gradually decreased. For the variant with 5% alternative raw material, WA dropped to 84.4%. This trend continued, reaching 78.8% with 25% alternative raw materials. At 50% alternative raw material, WA was 80.3%, and at 100% alternative raw material, it decreased to 72%. For the variant with 0% alternative raw material, water absorption after 24 h is 94.1%. As the proportion of alternative raw materials increases, water absorption decreases. At 5% alternative raw material, WA drops to 76.5%. This trend continues, with WA reaching 90.6% at 10% alternative raw material and 89.5% at 25%. At 50% alternative raw material, WA is 87.1%, and it slightly decreases to 85.7% at 100% alternative raw material. The clear varnish penetrated deeper into the wood (as shown in Figure 5), covering more of the wood surface. This deeper penetration likely contributed to the improved dimensional stability by reducing water absorption. The lacquer reduces the particles’ ability to absorb water, which in turn reduces the risk of swelling and weakening of the structure due to moisture exposure. White paint contains fillers that may be more susceptible to moisture than clear lacquer. A similar situation was obtained when testing particleboards made from other particleboards, where the WA also decreased as the proportion of recycled wood particles increased [47]. Factors influencing WA were also identified; these include the bulk density [66] of the starting raw material or internal bond quality; the weaker the WA, the more dense the bulk of the particleboard [67].

Figure 11 displays the results of thickness swelling after 2 and 24 h. As the proportion of recycled wood increases, the swelling decreases. After 2 h, for the variant with 0% alternative raw material, the thickness swelling was 35.2%. For the variant with 100% alternative raw material, TS increased to 39.3%. Thickness swelling after 24 h for the variant with 0% alternative raw material is 42.2%. As the proportion of alternative raw material increases, the thickness swelling decreases, reaching 30.8% with 100% alternative raw material (clear varnish). After 2 h, the thickness swelling for samples made from white-painted window frames was 35.2% for the variant with 0% alternative raw material. The introduction of alternative raw materials led to an increase in swelling, reaching 47.6% for the variant with 5% alternative raw material. Swelling then decreased with higher proportions of alternative raw material, reaching 43.1% at 10%, 36.8% at 25%, and 33.5% at 50%. For the variant with 100% alternative raw material, the thickness swelling was 28.9%. For the variant with 0% alternative raw material, thickness swelling after 24 h is 42.2%. The addition of alternative raw material initially reduces swelling, reaching 34.7% at 5%. However, TS increases to 45% at 10% and 25% alternative raw material and remains at 45% for 50% alternative raw material. At 100% alternative raw material, TS slightly decreases to 38.9%. This is attributed to the higher presence of varnishes in the manufactured panels, which are less prone to water absorption and, consequently, swelling. The best results were observed in samples with clear varnish, as they exhibited the lowest swelling. A similar trend was observed when testing particleboard made from recycled particleboard. The recycled boards were pre-cleaned by boiling, adequately prepared, and used to manufacture new particleboards [68].

### 3.7. Density Profile

Figure 12 and Figure 13 illustrate the density profiles of the manufactured particleboard variants produced from wood reclaimed from window frames. Upon analyzing the density profile of the panels made from white-painted window frame particles, it is evident that the overall reproducibility is relatively consistent, with no significant discrepancies. However, a closer inspection reveals a lower density in the central part of the panel. Additionally, density fluctuations become more pronounced as the proportion of recycled particles increases. These variations could contribute to the poor perpendicular tensile strength observed. Moreover, larger particles coated with varnish, which has a much higher density than wood, may also affect the density profile, leading to the observed irregularities in the graph. The particleboards’ density profile, made partially or entirely from particles derived from window frames painted with clear varnish, shows a similar trend to previous observations. However, a key difference is noted in the outer layers of the particleboards and their shape. This difference may be due to the order in which the layers were deposited, with finer particles settling at the bottom of the mats. Consequently, the density peak on the left side is more pronounced, particularly in the board where the content of recycled fraction is 100%. The brittleness of the varnish contributes to an increase in the proportion of the dusty fraction, affecting the overall density distribution. Research indicates that high-strength parameters can be retained if the core layers’ density can be reduced for various reasons—in this case, the presence of paint and varnish—while maintaining appropriate density in the surface layers [69]. The density of the tested particleboards in the core layer does not significantly differ from the reference variant. Still, it is worth noting that the bulk density of recycled materials was higher depending on the applied finish, which can also decrease the mechanical properties of the tested wood composites [70].

### 3.8. Formaldehyde and TVOC Emission

Total volatile organic compounds and formaldehyde emissions from transparent wood varnish and white paint, as presented in Figure 14, can exhibit similar levels after one year due to several factors. Initially, both products release a high concentration of VOCs and formaldehyde as solvents, and other chemical components evaporate during and after application. Over time, these emissions decrease significantly as the materials cure and stabilize. One year post-application, the residual emissions from both products primarily come from the degradation of the remaining organic compounds in the cured film. Transparent wood varnish and white paint often contain similar base components, such as resins, solvents, and additives, which release VOCs and formaldehyde as they slowly break down. The differences in initial composition become less significant over time as the more volatile components have evaporated, leaving compounds with similar stability and degradation rates behind. Moreover, environmental factors such as temperature [71], humidity [72], and air exchange rates affect the emission rates of both materials similarly. These factors influence the diffusion and off-gassing processes, leading to comparable TVOC and formaldehyde levels in indoor environments after prolonged exposure. Manufacturing standards and regulations also play a role. Both varnish and paint products are subject to stringent guidelines to limit harmful emissions, ensuring their long-term emission profiles align with safety standards [73]. Consequently, the residual levels of TVOCs and formaldehyde after a year tend to converge within a narrow range dictated by these regulations [74]. In summary, the similarity in TVOC and formaldehyde emissions from transparent wood varnish and white paint after one year is due to the comparable long-term behavior of their chemical components, influenced by environmental factors and regulatory standards. This convergence highlights the importance of considering initial and long-term emissions when evaluating coating products’ environmental impact and safety.

Formaldehyde and TVOC emission tests showed no significant differences in the manufactured panels compared to the reference variant. Formaldehyde emissions can depend on several factors, including ambient conditions such as temperature and high humidity. However, it is worth pointing out that the newly recycled window frames were manufactured using relatively new coatings, which may comply with current emission standards, or the formaldehyde content is negligible. A study comparing functionalized paint with regular formula paint found statistically significant decreases in formaldehyde levels in rooms coated with the functionalized paint. This emphasizes the critical role that paint quality plays in formaldehyde emissions. In addition, the recycled material was seasoned for about a year, so some of the formaldehyde may have escaped. 

## 4. Conclusions

The conclusions from the conducted study indicate that using recycled wooden window frames, painted with both clear varnish and white paint, as raw materials for particleboard production significantly affects their physical and mechanical properties. The results confirm that using recycled wood leads to substantial changes in particle structure and particleboard density, impacting strength parameters such as MOE, MOR, and SWR. An increase in the proportion of recycled wood particles increased the particleboards’ overall density and bending strength; however, not all particleboard variants met the MOR standard requirements. MOE values increased with the amount of recycled material. They met the standards, suggesting that recycled wood can be effectively used in particleboard production if the appropriate surface density is maintained. One of the key conclusions is also the influence of the type of coating on the particleboard properties. The clear varnish coating penetrates the wood better, leading to improved dimensional stability, reduced water absorption, and less swelling of the particleboards. In contrast, particles from frames painted with white paint had higher density and a greater proportion of fine fractions, which could negatively impact some mechanical parameters. The formaldehyde and total volatile organic compound emission tests showed that the emission levels in particleboards made from recycled wood were comparable to those of the reference material. This is likely because the applied coatings were relatively new and met current emission standards, as well as the long seasoning period of the recycled material, which may have led to a reduction in formaldehyde emissions.

In summary, the study’s findings suggest that recycled wood from window frames can be successfully used in particleboard production, provided that the production processes are adjusted to ensure the appropriate mechanical properties and dimensional stability of the particleboard, thus positively contributing to the circular economy policy. However, it is necessary to tailor the particleboard density individually depending on the type of raw material used.

## Figures and Tables

**Figure 1 materials-17-04132-f001:**
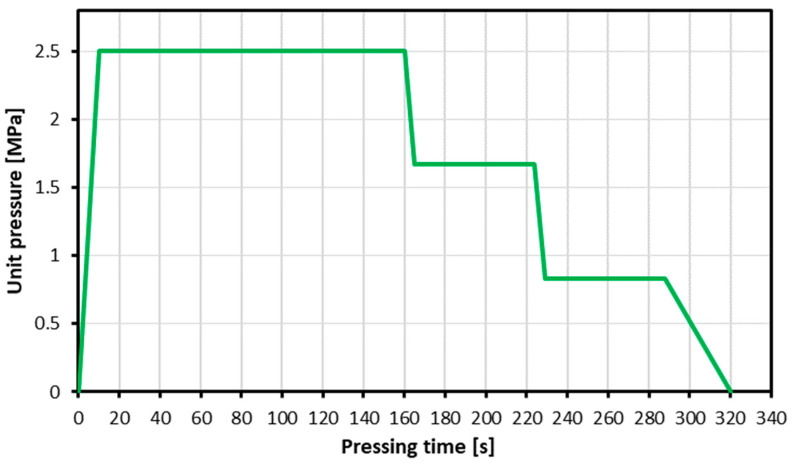
Particleboard pressing diagram.

**Figure 2 materials-17-04132-f002:**
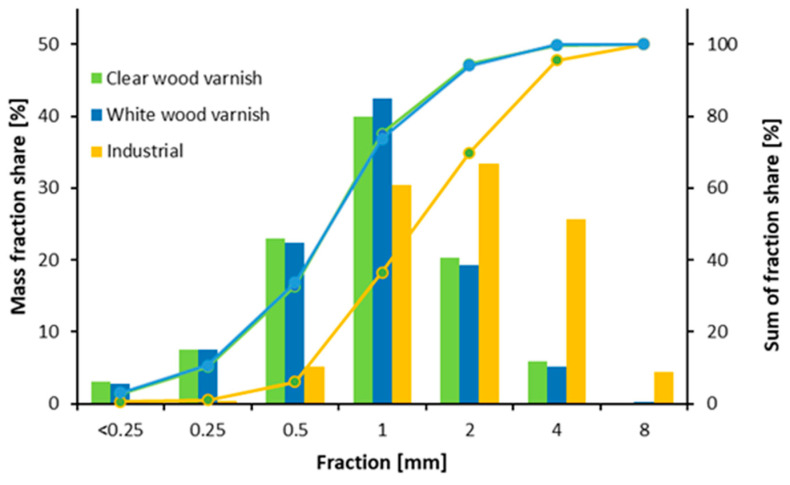
Mass fraction share of recovered particles used in research.

**Figure 3 materials-17-04132-f003:**
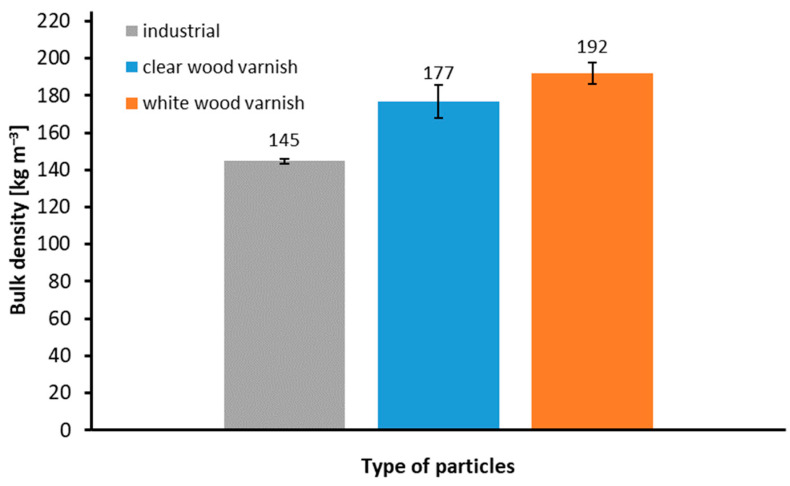
The bulk density of used particles.

**Figure 4 materials-17-04132-f004:**
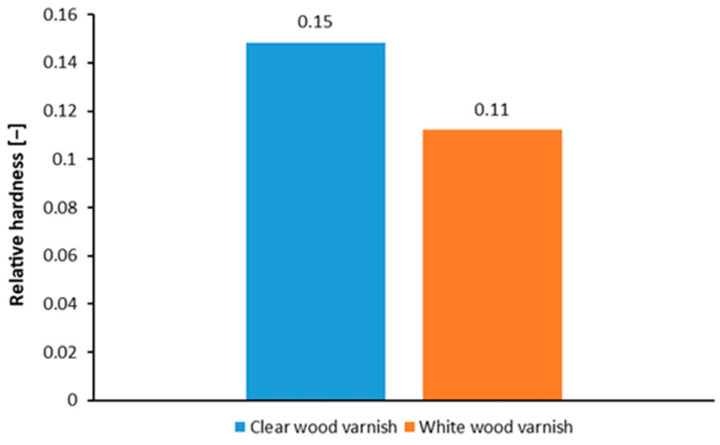
Relative hardness of the applied coatings.

**Figure 5 materials-17-04132-f005:**
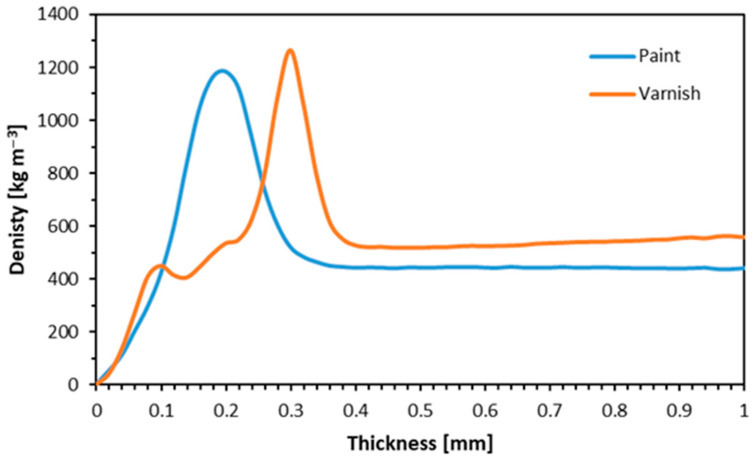
The density profile of the surface of the finished window frames.

**Figure 6 materials-17-04132-f006:**
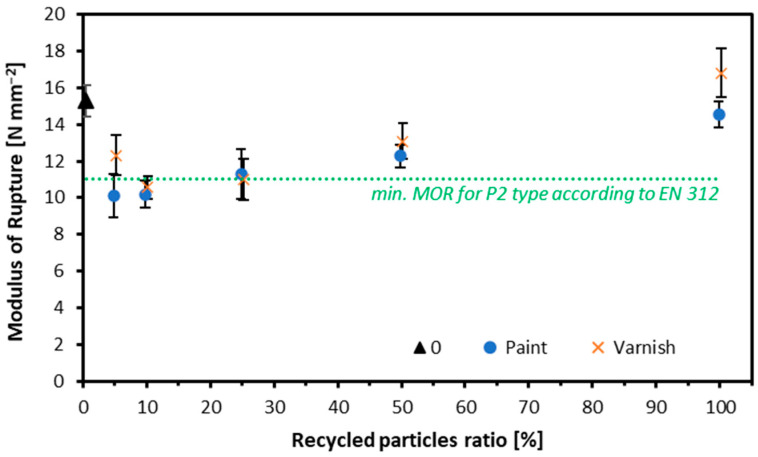
Modulus of rupture of tested composites.

**Figure 7 materials-17-04132-f007:**
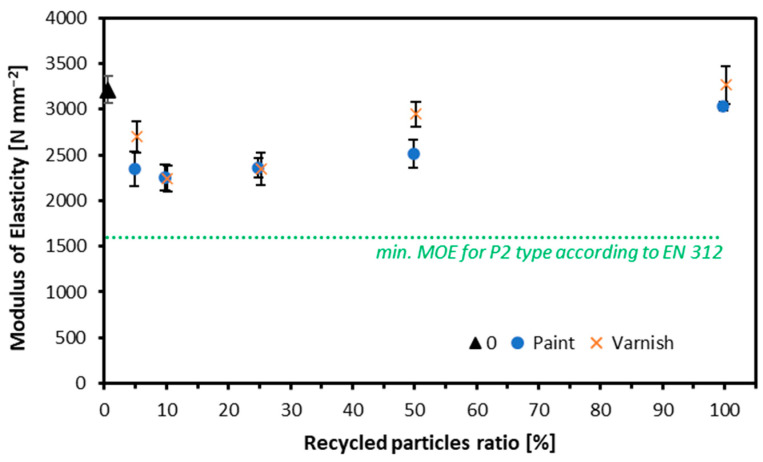
Modulus of elasticity of tested composites.

**Figure 8 materials-17-04132-f008:**
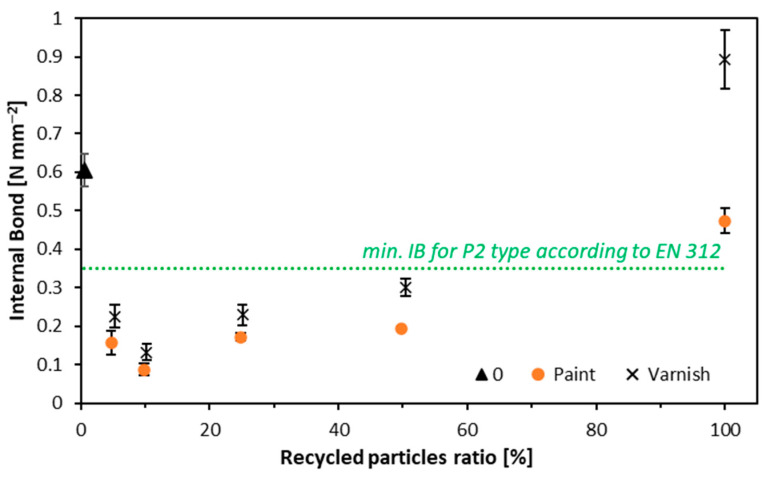
Internal bond of tested composites.

**Figure 9 materials-17-04132-f009:**
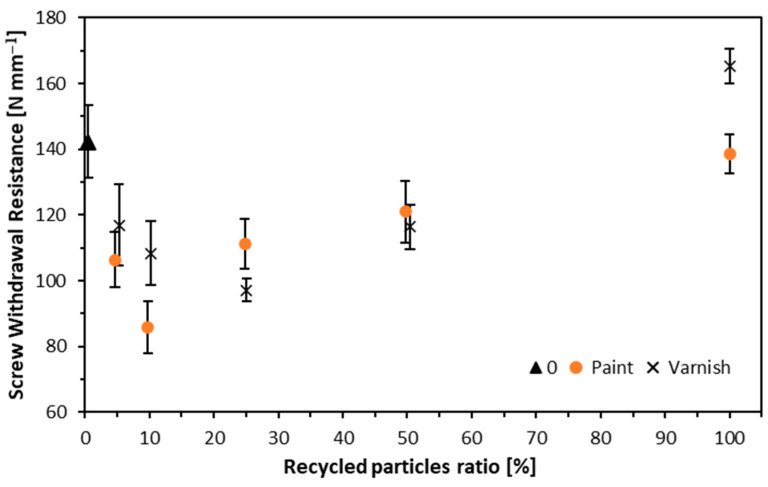
Screw withdrawal resistance of tested composites.

**Figure 10 materials-17-04132-f010:**
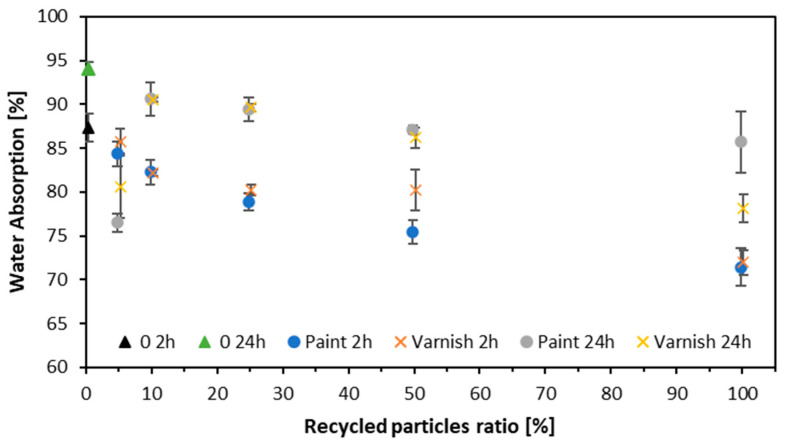
Water absorption of tested composites.

**Figure 11 materials-17-04132-f011:**
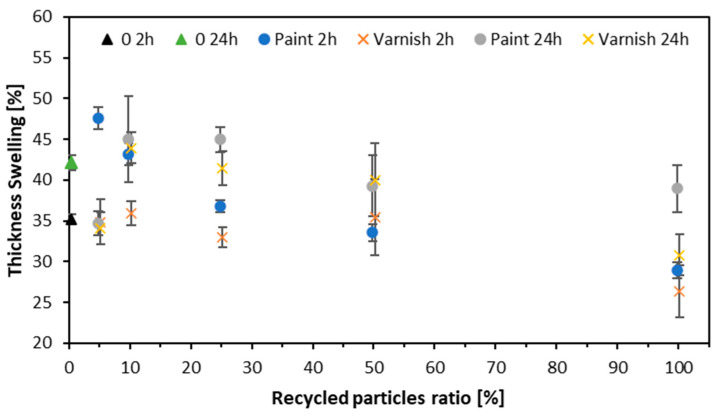
Thickness swelling of tested composites.

**Figure 12 materials-17-04132-f012:**
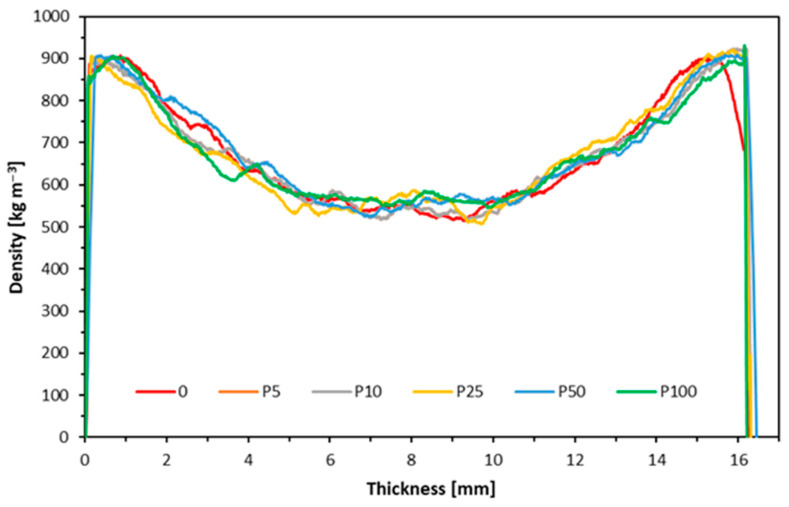
Density profile of composites made from recycled white-painted window frames.

**Figure 13 materials-17-04132-f013:**
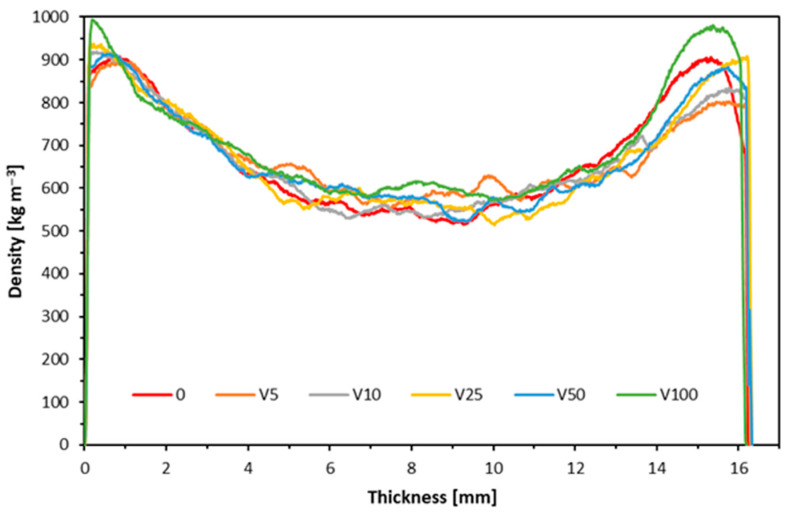
Density profile of recycled window frame composites painted with clear varnish.

**Figure 14 materials-17-04132-f014:**
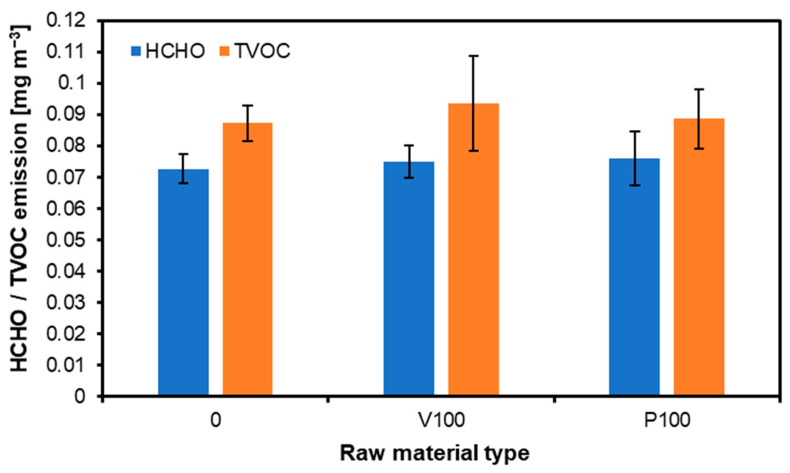
Formaldehyde and total volatile organic compound emissions of recycled window frame composites painted with various finishing materials.

**Table 1 materials-17-04132-t001:** The average, standard deviation, and statistical assessment results of mean values.

Test Type	Alternative Raw Material Particles Share [%]										
	0	5		10		25		50		100	
		V	P	V	P	V	P	V	P	V	P
MOE [N mm^−2^]	3212 ^a 1^ (146) ^2^	2697 ^b^ (169)	2344 ^c^ (186)	2241 ^c^ (140)	2251 ^c^ (138)	2350 ^c^ (177)	2357 ^c^ (108)	2949 ^b^ (136)	2511 ^c^ (158)	3265 ^a^ (207)	3032 ^b^ (43)
MOR [N mm^−2^]	15.3 ^a^ (0.9)	12.3 ^b^ (1.1)	10.1 ^c^ (1.2)	10.6 ^b^ (0.6)	10.2 ^b^ (0.7)	11.0 ^b,c^ (1.1)	11.3 ^b,c^ (1.3)	13.1 ^b^ (1.0)	12.3 ^b^ (0.6)	16.8 ^a^ (1.3)	14.5 ^a^ (0.7)
IB [N mm^−2^]	0.61 ^a^ (0.04)	0.23 ^b^ (0.03)	0.16 ^c^ (0.03)	0.13 ^c^ (0.02)	0.09 ^d^ (0.01)	0.23 ^b^ (0.03)	0.17 ^c^ (0.01)	0.30 ^e^ (0.02)	0.19 ^b^ (0.01)	0.89 ^f^ (0.08)	0.47 ^g^ (0.03)
SWR [N mm^−1^]	142 ^a^ (11)	117 ^b^ (12)	106 ^b^ (9)	108 ^b^ (10)	86 ^c^ (8)	97 ^b^ (4)	111 ^b^ (8)	116 ^b^ (7)	121 ^b^ (9)	116 ^d^ (7)	139 ^a^ (6)
TS 2 h [%]	35.2 ^a^ (0.6)	34.9 ^a^ (2.8)	47.6 ^a^ (1.4)	35.9 ^a^ (1.5)	43.1 ^b^ (1.3)	33.0 ^c^ (1.3)	36.8 ^a^ (0.7)	35.5 ^a^ (4.7)	33.5 ^b^ (1.0)	26.3 ^d^ (3.2)	28.9 ^d^ (1.0)
TS 24 h [%]	42.2 ^a^ (0.9)	34.1 ^b^ (1.9)	34.7 ^b^ (1.5)	43.9 ^a^ (1.9)	45.0 ^a^ (5.3)	41.5 ^a^ (2.1)	45.0 ^c^ (1.6)	40.0 ^a,b^ (4.5)	39.3 ^a,b^ (3.7)	30.8 ^d^ (2.5)	38.9 ^a,b^ (2.9)
WA 2 h [%]	87.4 ^a^ (1.6)	85.8 ^a,b^ (1.5)	84.4 ^b^ (1.5)	82.2 ^c^ (0.3)	82.3 ^b,c^ (1.4)	80.3 ^d^ (0.6)	78.8 ^d^ (1.0)	80.3 ^c^ (2.3)	75.5 ^e^ (1.4)	72.0 ^f^ (1.5)	71.4 ^f^ (2.2)
WA 24 h [%]	94.1 ^a^ (0.8)	80.6 ^b^ (3.6)	76.5 ^b^ (1.1)	90.5 ^c^ (0.2)	90.6 ^c^ (1.9)	89.6 ^c^ (0.4)	89.5 ^c^ (1.3)	86.2 ^d^ (1.2)	87.1 ^d^ (0.5)	78.2 ^b^ (1.6)	85.7 ^d^ (3.5)
HCHO [mg m^−3^]	0.073 ^a^ (0.005)	n.b.t. ^3^	n.b.t.	n.b.t.	n.b.t.	n.b.t.	n.b.t.	n.b.t.	n.b.t.	0.075 ^a^ (0.005)	0.076 ^a^ (0.009)
TVOC [mg m^−3^]	0.087 ^a^ (0.006)	n.b.t.	n.b.t.	n.b.t.	n.b.t.	n.b.t.	n.b.t.	n.b.t.	n.b.t.	0.094 ^a^ (0.015)	0.089 ^a^ (0.010)

^1 a–f^ homogeneous group. ^2^ standard deviation (in brackets). ^3^ n.b.t.—not being tested.

## Data Availability

https://doi.org/10.18150/WNCGBJ (created and accessed on 19 July 2024).

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
