# Peer review of "Supporting Circular Economy Principles by Recycling Window Frames into Particleboard"

_materials, 2024, doi:10.3390/ma17164132_

Round 1

Reviewer 1 Report

Comments and Suggestions for Authors

The article analysis particleboards from recycled window frames. The thematic of the article is of great importance due to support of circular economy. Even though the aim of the current study is very good, the results presentation lacks quality. Therefore, I have few suggestions to improve the quality of the current study:

1. I would suggest not use such terms as "garbage" or "trash". Please use instead "waste" as it is more scientific term.

2. Introduction section lacks information of what has been already achieved by other authors in the same field, what has to be done to overcome the drawbacks which were analysed in previous scientific works. In current state, the introduction section is a bit abstracts and does not say anything about the relevance or novelty of the current work.

3. Materials section misses the information about the company which provided the industrial particles.

4. Preparation of Panels section does not provide any information about the glue used.

5. Wronka and Kowaluk (2022a) in page 5, line 211 does not have a citation reference in the brackets.

6. The quality of the figures should be increased.

7. The results and Discussion section is quite weak as it does not almost no numerical values which could indicate the magnitude of increment of decrease in values obtained from various tests. Also, the in depth analysis on why one or another differences between the numerical values are obtained are missing.

8. Structural analysis of the composites would supplement the article in regards of quality. The analysis of the relation between the structure and strength parameters is mandatory in my opinion.

Author Response

Dear Reviewer, I've attached our response to your review. Kind regards!

Reviewer 2 Report

Comments and Suggestions for Authors

This paper deals with the possibility of using recycled wood of window frames for particleboards production. The window frames were experimentally produced and before being recycled into particleboards, were used in realistic conditions for one year. The window frames were produced using two types of surface coatings - clear and white wood varnish.

The manuscript is somewhat well prepared but there are few details that must be addressed. The first of which is the title, where circular economy is mentioned, but in no part of the manuscript (exerting Introduction) is the circular economy even mentioned. This is most obvious in the Results part of the manuscript where not a single word or explanation of the obtained results are given in that direction. This must clearly be dealt with either by changing the title of the manuscript, or by giving additional comments on the obtained results. Here I must also say that the Conclusions part given in lines 409-420 is clearly not related to the research and must be omitted. Also, question that have not been answered in the manuscript is – Why weren’t “real” refurbished window frames used for particleboards? By term “real” I mean those that were withdrawn from the use after several (decades) years. If it is because the Authors wanted to control the raw material from the start, then this must be clearly stated in the manuscript. However, the question arises whether this is a good approach, as the results are somewhat defined already at the beginning, by controlling the experiment right from the window frame manufacturing. Therefore, in my opinion the results given in this manuscript are just barely usable in terms of using wood withdrawn from usage for particleboards manufacturing on industrial scale.

As for more specific remarks, the knife mill description is almost identical in lines 136-141 and lines 163-167, so one (latter) of those descriptions should be omitted. To give the pressing diagram would have been much better than the description given in lines 153-158. The same remark goes also on the results presentation where a table(s) with the results of examined physical and mechanical properties of particleboards alongside the statistical analysis (as in Table 1), would have been more appropriate than the graphical representations on Figures 5-10. At the beginning of the chapter 2.3. Characterization of panels, the Authors mention that the coating hardness and granulometric composition of wood chips were determined, but those parameters can’t be considered as a part of panels characterization. Therefore, I suggest to add another subsection where the methodology of determination of those properties are given. As my final remark, I must say that the Authors cited more than a few of their earlier works and that the mentions of those papers weren’t necessary in terms of uplifting the overall quality of the manuscript.

Comments on the Quality of English Language

The quality of the English language in which the manuscript is written is fine, as there are only few errors that need to be corrected at proofreading.

Author Response

(The authors gave the same response as above.)

Reviewer 3 Report

Comments and Suggestions for Authors

This article is more like a commodity technical report to me than a research article. The research focus on utilizing shredded wood window frames of the two most common surface finishes to create three layers of particleboard and an assessment of the factors limiting that way wood reuse, but it kind of lacking the novelty of a research that should be.

I would suggest journal to reconsider accepting this article for publication.

Author Response

(The authors gave the same response as above.)

Round 2

Reviewer 1 Report

Comments and Suggestions for Authors

Authors have taken into consideration all my remarks. Thank you. Therefore, the article can be accepted.

Author Response

Dear Reviewer, thank you for your valuable remarks and thanks for the acceptance of our contribution to manuscript revision!

Reviewer 2 Report

Comments and Suggestions for Authors

Dear Authors,

I have carefully read the revised manuscript, and must say that I'm glad that the majority of my recommendations were accepted. Still, I have found some details that must/should be addressed before the manuscript is published. More precisely, in lines 209-210 the correct way for giving a citation must be used (currently it states: "Error! Reference source not found".), and text break between lines 448 and 449 must be omitted. Also, I would recommend that the sentence in lines 155-157 is re-written in such way that you do not provide the name of the coatings manufacturer, and simply state that coatings from renowned manufacturer were used. Later in the manuscript Sigma Coatings are once again mentioned, and this part should also be re-written. This is due to the fact that the topic of this research weren't coatings and that some results on coatings were given. In order not to give positive/negative review of the coatings themselves and their manufacturer, I would strongly recommend to correct the text. 

Kind regards,

Reviewer

Author Response

(The authors gave the same response as above.)

Reviewer 3 Report

Comments and Suggestions for Authors

Thank you for the reply. After read through the authors' response, I understand this study would motivate the recycling of wooden building, especially in implementing European Union guidelines on building thermal modernization. This article should be ready for publication.

Author Response

Dear Reviewer, thank you for your valuable remarks and thanks for the acceptance of our contribution to manuscript revision!

Yes, in our research, we tried to highlight and pay attention to the existing building modification processes and give the proven example of wood recycling, which is also in line with European and global policies.

Thank you for understanding!